# Coupled Response of Membrane Hydration with Oscillating Metabolism in Live Cells: An Alternative Way to Modulate Structural Aspects of Biological Membranes?

**DOI:** 10.3390/biom9110687

**Published:** 2019-11-02

**Authors:** Luis A. Bagatolli, Roberto P. Stock, Lars F. Olsen

**Affiliations:** 1Instituto de Investigación Médica Mercedes y Martín Ferreyra—INIMEC (CONICET)—Universidad Nacional de Córdoba, Friuli 2434, Córdoba 5016, Argentina; 2Departamento de Química Biológica Ranwel Caputto, Facultad de Ciencias Químicas, Universidad Nacional de Córdoba, Córdoba 5000, Argentina; 3MEMPHYS—International and Interdisciplinary Research Network, 5230 Odense, Denmark; rpstepoz@gmail.com; 4University of Southern Denmark, Institute for Biochemistry and Molecular Biology, Campusvej 55, 5230 Odense, Denmark; lfo@bmb.sdu.dk

**Keywords:** water activity, crowding, lyotropic mesomorphism, biological membranes, mesophases, ATP, cytoskeletal proteins, 6-acyl-2-(dimethylamino)naphtalenes fluorescence probes, association-induction hypothesis (AIH)

## Abstract

We propose that active metabolic processes may regulate structural changes in biological membranes via the physical state of cell water. This proposition is based on recent results obtained from our group in yeast cells displaying glycolytic oscillations, where we demonstrated that there is a tight coupling between the oscillatory behavior of glycolytic metabolites (ATP, NADH) and the extent of the dipolar relaxation of intracellular water, which oscillates synchronously. The mechanism we suggest involves the active participation of a polarized intracellular water network whose degree of polarization is dynamically modulated by temporal ATP fluctuations caused by metabolism with intervention of a functional cytoskeleton, as conceived in the long overlooked association-induction hypothesis (AIH) of Gilbert Ling. Our results show that the polarized state of intracellular water can be propagated from the cytosol to regions containing membranes. Since changes in the extent of the polarization of water impinge on its chemical activity, we hypothesize that metabolism dynamically controls the local structure of cellular membranes via lyotropic effects. This hypothesis offers an alternative way to interpret membrane related phenomena (e.g., changes in local curvature pertinent to endo/exocytosis or dynamical changes in membranous organelle structure, among others) by integrating relevant but mostly overlooked physicochemical characteristics of the cellular milieu.

## 1. Background

In the last five years, we have studied distinct physicochemical aspects of a well-known biological oscillator, namely, oscillating glycolysis, in the yeast *Saccharomyces cerevisiae*. When these cells are starved and respiration is blocked, they will use externally added glucose in such a way that the chemical activity of essential central metabolites such as reduced Nicotinamide Adenine dinucleotide (NADH) and Adenosine Triphosphate (ATP) will follow a well-established temporal waveform. This remarkable oscillatory behavior was discovered over 60 years ago [1] and has been studied intensely in both intact cells and cell extracts [2,3]. While the circumstances under which glycolytic oscillations take place are documented and the enzymes and participating biochemical reactions have been characterized, an adequate explanatory framework remains unspecified: why glycolytic oscillations arise and what they reveal about the complex relationship between the various levels of the description of life processes are questions still awaiting definite answers.

The classical approach to comprehend why glycolysis oscillates envisages the pathway largely in isolation from the rest of the cell. Particularly, it considers the intracellular environment as an isotropic aqueous solution where enzymes and metabolites react and diffuse according to principles of mass action kinetics and dilute solution theory (van’t Hoff). Although this interpretation exerts a robust influence on the way we currently grasp and think about biological systems, it has been repeatedly challenged when consideration is given to well-accepted (although largely ignored) features of the cellular milieu such as molecular crowding and spatial confinement that are likely to strongly influence the way that the cellular machinery operates. In this respect, it is important to underscore the strong influence of these environmental features on the most abundant component of biological systems (i.e., water). Due to the high concentration of macromolecules, estimated to be around 200–350 mg/mL [4,5], the cell interior is a very crowded environment. Therefore, it is unlikely that it will share properties of dilute solutions where enzymes are generally characterized; in fact, it is becoming increasingly evident that intracellular water is affected by such a high concentration of solutes and that its behavior significantly deviates from tractable ideality. Furthermore, water in the cell is not just a passive medium where reactions occur, but an active participant (e.g., many cellular reactions are condensations that produce water, or hydrolytic ones that consume it). Water also participates as a key thermodynamically active structural component of lipid membranes [6,7], playing an important role in their topology and function. However, *the role of water as an active component* is seldom acknowledged in the current models describing the cell and, as a consequence, in those defining biological membranes. Finally, it is important to remark that there is still an unresolved debate about the state of water in intact cells. While some suggest that most intracellular water is in a state akin to dilute solutions [8], others claim that it exists in a H-bonded polymer-based gel-like state [9,10,11].

In a series of papers [12,13,14,15], we reported a number of key observations that challenge some basic tenets of the established view of cellular systems. Particularly, we observed that during glycolytic oscillations, the extent of dipolar relaxation in the cellular interior, which is governed mainly by the rotational dynamics of intracellular water, is coupled to the behavior of the pathway (intervening metabolites such as ATP and NADH) in that it oscillates synchronously [12,13]. We showed that this coupling is obliterated by inhibiting ATP production and that it depends on a functional actin network [13]. We also found that intracellular water relaxation is a crucial parameter for the manifestation of glycolytic oscillations (e.g., oscillations occur only within a defined range of relaxation values), and this parameter emerges as an intensive property of the system (i.e., it is scale independent) [12,13]. These observations call into question the idea that intracellular water acts as a mere passive biological solvent, as implicitly understood in the standard membrane-based model of the cell. In a subsequent study, we observed that cell volume, heat flux, and temperature also oscillated synchronously with glycolytic oscillations, showing a strong coupling of these physical properties with the biochemical pathway [14]. These results were consistent with a recently proposed thermodynamic formalism where isentropic thermodynamic systems can display coupled oscillations in all extensive and intensive variables, reminiscent of adiabatic waves [16]. This approximation suggests that glycolytic oscillations may arise as a consequence of a living cell’s requirement for a near constant low-entropy state while remaining metabolically active, in contrast to the prevailing notion that metabolic processes have to be understood as essentially dissipative (i.e., in terms of steady state energetics).

Taken together, these results support the view that the general principles of coupling are at play and that glycolysis as an enzymatic pathway both influences and is influenced by physical properties of the cellular milieu that are cyclically altered by the cyclic accumulation/disappearance of the products of the (metabolic) pathway. This is in stark contrast to those models that seek to understand glycolysis (and its oscillations) purely in terms of the kinetic properties of the enzymes involved, as characterized in classical dilute systems (i.e., where water activity is very high). They also support the view of the cellular interior as a highly structured and near equilibrium system where energy inputs can be low and sustain regular oscillatory regimes. We argue that all of these experimental results [12,13,14,15], which are difficult to rationalize under the dominant view of the cell, are in line with the association-induction hypothesis (AIH) [17,18,19]. This theory, which still remains unrefuted, offers an alternative conceptual approach based on well-established principles of colloidal physical chemistry to explain how emergent features of the intracellular environment may give rise to cellular function.

## 2. Brief Account of Ling’s Association-Induction Hypothesis

Based on concepts derived from the physical chemistry of colloids, the AIH is a general theory of cell physiology. Introduced by G.N. Ling in 1962, the AIH developed a rigorous statistical mechanical theoretical scaffold, along with extensive experimental testing, to establish general principles of cell-wide coupling [17,18,19]. Conceptually, it relies on two components: (i) a revision of the behavior of ionizable species in environments where ionic strength is so high (crowded) that classical Debye-Hückel approximations are rendered essentially useless; and (ii) a theory of the dynamics of the structure of proteins based on their characteristics as both polyelectrolytes and partially resonating polymers. The first component consists of an in-depth analysis of ion dissociation from the standpoint of Bjerrum’s concept of a non-dissociated fraction. Briefly, it demonstrates that in situations where there is a system of charges with limited mobility (such as those on proteins or in ion-exchange resins) the activity of free (dissociated) counter ions (of higher mobility such as, but not limited to, H^+^, Na^+^ or K^+^) is drastically reduced in comparison to dilute solutions. This analysis provides the “association” component of the AIH. The second component is a theory of the behavior of proteins in terms of their nature as polyelectrolytes, which extends the concept of ionic association summarized above to its effects on high molecular weight molecules bearing multiple charged sites linked by a partially resonant backbone. The AIH shows that, as polyelectrolytes, the charged groups on proteins must exist in mostly associated states, either to other charges in the protein (salt bridges) or to counterions. Furthermore, since ionic dissociation is greatly impeded by the high amount of (fixed) charges within an intracellular environment, most ionic behavior involves ion exchange (as opposed to simple Debye-Hückel dissociation). Consequently, the AIH develops a measure of the relative affinity of monovalent ions for ionizable sites on protein chains (the “c-value”) [17,18,19]. A key insight of the AIH is that these *ion exchanges involve energy flows that exert local effects that are propagated along the resonant polypeptide backbone of proteins*. The propagation of these ionic associations along the semi-resonant backbone of proteins dynamically alters both their relative affinity for other charges/counter ions in their milieu and their structural characteristics, mostly in terms of H-bonding to the surrounding water and/or other proteins. All this results in the dynamic long-range transfer of information and energy via conformational changes (for example, α-helix to random coil transitions) or subunit dissociation, providing a more dynamic view of protein structure. This component provides the “induction” part of the AIH and it is further refined into a profound theory of cooperativity and emergent properties central to living phenomena. From these two components of the AIH (i.e., association and induction) an integrated theory of cellular activity emerges. Unlike the mainstream view, the theory quantitatively approaches the living state as a metastable near-equilibrium system that requires relatively small energy inputs to maintain its highly constrained dynamical properties.

Remarkably, in light of the AIH, the “induction” part above-mentioned is regulated by key cellular ubiquitous solutes called cardinal adsorbents. These cardinal adsorbents (for example, ATP, hormones, divalent ions) *modulate intrinsic inductive properties of many intracellular proteins upon association* via polarizing effects. In this context, and based on extensive experimental work, Ling demonstrated that the centrality of ATP (the most prominent cardinal adsorbent) in cellular activity resides in its ability to control the bulk properties of the cell interior, as an amalgamated water-protein-ion/solute system, via this mechanism [17,18,19].

Unlike mostly static protein conformations that favor intramolecular electronic interactions, such as α-helix or β-sheet (which dominate the structure of globular proteins; called introverted systems in the AIH), the AIH assigns great importance to the random coil (or extended) conformations, which promote potential electronic interactions with the surrounding environment. Since extended conformations expose carboxyl groups of charged amino acids and backbone imino and carbonyl groups involved in the highly polarizable amide bond, interactions with monovalent ions (e.g., K^+^, Na^+^) and intracellular water are favored [18,19]. In the AIH, a protein exhibiting an extended state is called an extroverted system, in other words, more likely to H-bond and polarize the surrounding water, altering its properties (such as its relaxation dynamics, see below). Examples are fibrillar proteins (such as actin and tubulin) or proteins displaying a high contribution of random coil conformations, now known as intrinsically disordered proteins [20].

Concerning the state of water in the cell, the AIH incorporates a subsidiary theory called polarized-oriented multilayer theory of cell water [19,21]. It develops the idea that protein backbones, through the carbonyl and imino groups of the amide bond, offer properly-spaced alternating negative–positive (called N–P) sites that orient and polarize a layer of oppositely-oriented water dipoles that in turn adsorbs, orients, and polarizes another layer of oppositely-oriented water dipoles, and so on. This constitutes a dynamic structure of multiple layers of polarized-oriented water dipoles that is cooperatively modulated by metabolic activity (e.g., levels of ATP) [19,21]. Specifically, the polarized state of intracellular water will be highly favored when ATP levels are high, and it will be reverted by decreasing intracellular levels of this metabolite, since it acts as a cardinal adsorbent and can induce structural changes in proteins and promoting transitions to extroverted conformations [18,19,21]. Importantly, the dynamic structure of polarized water dipoles will cause a reduction in the translational as well as rotational motional freedom of the water molecules and, more importantly, *it will influence the water chemical activity* [19]. This has been substantiated by a large amount of experimental work performed in both in vitro and in vivo systems, suggesting that most of the intracellular water exists in a polarized state when the cell is in a resting state [18,19]. This view offers a reasonable explanation of our results on the oscillatory coupling between intracellular water relaxation (measured by the fluorescence response of a series of 6-acyl-2-(dimethylamino)naphtalene probes) and central metabolites (such as ATP) during glycolytic oscillations including its dependence on the integrity of the actin cytoskeleton [12,13,15]. For instance, using a mathematical model based on the Yang-Ling isotherm [22,23], which is grounded on the principles of the AIH, we were able to adequately reproduce the coupling between the oscillations of ATP and the intracellular water dynamics observed in our experiments [13,15]. We also found that the Yang-Ling isotherm was more reliable in describing the kinetics of glycolytic enzymes in crowded environments than classical models based on mass action kinetics such as Michaelis-Menten (MM) and Monod-Changeux-Wyman (MCW) [15]. The Yang-Ling isotherm is a general framework for models in the sense that it can generate both hyperbolic (MM) and sigmoidal (MCW) kinetics for single or multiple subunit enzymes [15].

To conclude this section, the AIH hypothesis allocates a key role to the cell interior (cytosol/organelles) as a responsive amalgamated water–protein–ion/solute system to explain cellular function. However, it does not assign a role to lipid structures other than to generically stabilize the protein–water–ion/solute system existing at the cell (or organelle) surfaces [19]. In the following sections we will, based on recent data, discuss how other dynamical/structural aspects of lipid structures could be incorporated into the AIH.

## 3. Relevant Results Concerning Membranous Systems

Of marked importance for the content of this article was the tight coupling observed between intracellular ATP levels and the dynamic state of intracellular water [12,13,15]. As briefly described above, we exploited the fluorescence response of a series of 6-acyl-2-(dimethylamino)naphtalene probes exhibiting different affinities for hydrophobic/hydrophilic environments, namely 6-acetyl-2-(dimethylamino)naphtalene (ACDAN), 6-propionyl-2-(dimethylamino)naphtalene (PRODAN), and 6-lauroyl-2-(dimethylamino)naphtalene (LAURDAN) (generically called DAN probes) to measure this phenomenon [12]. Following the original idea of G. Weber, who synthesized these fluorescent compounds to study the “relaxation phenomena in biological environments” [24], we generalized the well characterized mechanism of these fluorescent molecules (particularly of LAURDAN and PRODAN in membranes, which are responsive to the rotational dynamics of water in the probe’s milieu [25]), to study water dipolar relaxation throughout the interior of the cells [26].

Oscillations in the fluorescence response of two of the DAN probes (ACDAN, the most hydrophillic, and LAURDAN, the most lipophillic), and of NADH and ATP during glycolytic oscillations are summarized in Figure 1A. This figure also includes the fluorescence response of 9-diethylamino-5-benzo[a]phenoxazinone (Nile red), a lipophilic fluorescence reporter that partitions to lipid droplets in cells. Similar to the DAN probes, Nile red displays solvatochromic properties and responds to water relaxation [27,28]. A hallmark of the coupling between all of these oscillations is that they displayed the same frequency, as shown in the corresponding power spectra (Figure 1B). Of particular interest is the observation that the fluorescence emission of both LAURDAN and Nile red membrane probes oscillate just like the more water-soluble probe ACDAN (and PRODAN [12]), supporting the notion that the coupling is cell-wide (i.e., it is observed in both aqueous environments and lipid structures) [12,13,15].

As succinctly mentioned in Section 2, we constructed a mathematical model based on the AIH [13,15]. This new model, which accounts for our experimental results, allows for a chemo-electrodynamic coupling of the biochemical reactions that participate in glycolysis to the *physical* state of intracellular water. Briefly, the model places phosphofructokinase at the center of the metabolic oscillations, assuming that the ATP produced is adsorbed to the intracellular proteins (e.g., actin) polarizing intracellular water [13,15]. This model successfully explains why oscillations appear to be cell-wide and most cellular regions oscillate with the same frequency. In addition, given the obvious constraints imposed by a crowded environment to diffusion, a mechano-chemical coupling via structured or polarized water—controlled by the association of ATP to favor extended protein conformations—can also account for the almost instantaneous spatiotemporal coupling of the oscillations.

Notably, we showed that this new model also predicts a spatiotemporal coupling in the polarized state of water from cytosolic cellular regions where glycolytic oscillations occur (p) to regions where glycolysis is not occurring (p_1_) (i.e., membranes). The telling property of this behavior is that the oscillations observed in these different regions show the same frequency, but are slightly phase shifted. A theoretical result that assumes a simple mechanical coupling from p to p_1_ is shown in the phase plot included in Figure 2A [13]. We experimentally tested this prediction of the model by comparing the temporal fluorescence response of the lipophilic dye Nile red with those of the DAN probes. As shown in Figure 1 and [12,13,15], Nile red and the three DAN probes oscillate at the same frequency. However, the phase analysis of the experimental data (Figure 2B) indicates that while Nile red and LAURDAN oscillate in phase with each other, a progressive phase shift is observed during the oscillations from PRODAN to ACDAN [15]. This trend follows the relative affinity of the probes for hydrophilic/hydrophobic environments (ACDAN being the most hydrophilic of the DAN series), and seems to verify the prediction of our model [13,15]. We conject that the oscillatory behavior of Nile red (and LAURDAN) arises as a consequence of the coupling of water associated with the polar groups of lipids to the oscillating polarized state of water in the cytosol, where glycolysis primarily occurs. In summary, what this finding reveals is that *the polarized state of water in the cell is transmitted from the cytoplasm to water existing in membranous structures*.

At this point, it is interesting to comment on our interpretation of the data obtained with the more hydrophobic DAN probe (i.e., the well characterized fluorescent probe LAURDAN), which shows a preferential partition to membranes and is insoluble in water [25,29]. It is well understood that the fluorescent response of this probe in membranous systems relates changes in the lateral structure of lipids (e.g., order parameter) to changes in the degree of water dipolar relaxation at the membrane interface [25,29,30]. Therefore, the oscillations observed in LAURDAN fluorescence during oscillatory metabolism (Figure 1, see also [12]) can be interpreted as the presence of periodic (local) structural changes of the lipid component of biological membranes originating from the transmission of the polarized state of water from the cytosol, in concordance with our model.

This idea also extends to the behavior of other probes that partition to membranous structures such as the membrane potential sensitive dye Mitotracker^TM^ red CMXRos (Mitotracker red). This cationic dye is believed to locate almost exclusively to the mitochondria, and its fluorescence response is considered to be a measure of the electrical potential at the mitochondrial membrane [31]. However, in micromolar concentrations in yeast, this dye stains many lipid structures other than mitochondria (Figure 3A), and more importantly, its fluorescence intensity oscillates in synchrony with glycolysis (Figure 3B). The power spectra of the oscillations of NADH and Mitotracker red are shown in Figure 3C, showing that (*i*) they are coupled and (*ii*) Mitotracker red oscillates at the same frequency as all the other variables shown in Figure 1.

## 4. Impact of These Observations on the Current View of Biological Membranes

The experimental evidence obtained so far by our group [12,13,14,15] puts into consideration an important paradigm shift concerning the role of water in cellular processes. It implies a drastic change in the view of the cellular interior, traditionally seen as an isotropic aqueous environment where mass action kinetics and van’t Hoff dilute solution theory apply, to a highly dynamic and structured milieu. This warrants an additional question: how can the dominant model of the cell explain the behavior observed in Figure 1, Figure 2 and Figure 3 if glycolysis is assumed to occur in the cell cytoplasm dominated by the presence of (normal and passive) liquid water and without the intervention of membranous structures? In this context, the AIH has emerged as a suitable theory to address the observed phenomenon, particularly since it incorporates the idea that the activity of water can be (either globally or locally) modulated by metabolism. A key issue here is that changes in water activity can easily exert a variety of structural changes in membranous structures, as discussed below.

It is well known that lipids display a rich polymorphism upon self-aggregation, and that this behavior can be controlled either by changes in temperature (*thermotropic* mesomorphism) or the activity of water (*lyotropic* mesomorphism). In the latter case, it is well established that changes in hydration modify lipid geometry and built-in curvature stress in lipid bilayers causing, under some conditions, transitions among different mesophases (e.g., from lamellar to non-lamellar phases) [32]. A very large body of studies characterizing various lipid mesophases in lipid/water systems (e.g., hexagonal, cubic, micellar) has been reported in the literature and discussed in terms of their relevance in biological systems since the early 60s [33,34,35,36]. However, with some exceptions, a number of mesophases produced by lipid self-aggregation have been disregarded and are not considered to be “biologically relevant”, mainly because of their low structural stability under conditions of high water activity [36], giving a strong prominence to (passive) lamellar structures (or bilayers). The argument rests on the general assumption that high water activity dominates the scene in cellular systems. In fact, some relevance is (quite reasonably) ascribed to non-lamellar mesophases only in particular cases such as those of anhydrobiotic organisms [37].

Considering the experimental results obtained by our group [12,13,14,15] and the interpretative framework provided by the AIH, we propose that active metabolic changes, reflected, for example, in the intracellular activity of ATP, regulate (local or global) dynamical changes in the structure of biological membranes via the physical state of cell water. This regulation may underlie physical cellular phenomena where membranous lipid structures are actively involved (e.g., changes in curvature during endo/exocytosis or organelle dynamics (mitochondrial fusion, [38]), generation of solitary pulses in axonal membranes [39], and other changes in the cell involving membranous structures). Interestingly, most of the processes indicated above involve the (direct or indirect) participation of key components considered in the AIH such as relevant metabolites such as ATP (or GTP) and fibrillar (cytoskeleton) proteins (which, for example, have ATP- or GTP-ase activity) or proteins containing a large contribution of random coil conformations (i.e., intrinsically disordered proteins), which are abundant and are involved in important regulatory processes in eukaryotes [20,40]. Additionally, possible changes in the chemical potential of intracellular water regulated by metabolism can straightforwardly account for modulations in the activity of lipases near membranous structures [41], or act directly in modifying the hydration of distinct lipid species that in turn can affect the membrane’s curvature stress field [32]. For instance, the role of lipid polymorphism in local events such as membrane fusion have already been proposed in the “metamorphic mosaic” model [35,42], however, an active role of intracellular water is simply ignored [6]. This phenomenon may also have an important influence on the regulation of melting transitions as already observed in native biological membranes [43] including its influence on membrane mechanical properties and the generation of lipid domains. A sketch of our hypothesis is included in Figure 4.

## 5. Concluding Remarks and Future Perspectives

Based on the study of the response of some membrane fluorescent probes during oscillatory glycolysis, we propose that metabolic changes may regulate (locally or globally) dynamic changes in the structure of biological lipid assemblies via the physical state of cell water, challenging the view that water acts mostly as a passive medium in the cell. This proposal is consistent with the association-induction hypothesis, which was originally introduced by G. Ling in 1962 [17] and further elaborated later on [18,19]. Nevertheless, the AIH debates that lipids form a continuous barrier in cellular environments, limiting their role to stabilizing the protein–water–solute system existing at the cell or organelle surfaces [19]. In other words, the AIH does not explicitly consider any regulatory effect of the physical state of intracellular water on structural (and dynamic) aspects of lipid structures, as proposed in this article. Therefore, one of the important challenges to test our (new) proposition is to design appropriate artificial model systems incorporating relevant physicochemical features of the cell milieu such as crowding and confinement.

The use of artificial membrane models, generally composed of pure lipids (although in some cases with proteins incorporated under controlled conditions), to mimic biological membranes is well accepted. Native membranes isolated from cells have also been used to overcome concerns connected to the lack of compositional complexity in these models. However, the vast majority of models (if not all) are prepared under conditions of excess water (very high water activity), precluding a systematic evaluation of the impact of key intracellular features such as crowding or spatial confinement. Although simple membrane models (for example, the classical liposomes) have historically contributed to an understanding of important aspects of membranous structures, it remains unclear in our opinion whether they can adequately model what happens in cellular environments. It is important to mention, however, that some attempts have been made at assembling experimental models such as the so called active membranes [44] that endeavor to capture relevant features of their biological counterparts. These encompass active enzymatic processes (such as vesicles containing Na^+^-K^+^ ATPase [45]), protein photo activation (bacteriorhodopsin, [46]), coupling membranes with proteins participating in the cell cytoskeleton [47,48], or evaluation of the influence of extroverted polymers on membranes by incorporating them in the lumen of giant vesicles [49]. Still, the potential effects on water polarization as described in the AIH throughout the polarized-oriented multilayer theory of cell water [19,21] (see Section 2) have not been adequately scrutinized in these systems; water continues to be seen as a passive/isotropic medium both in artificial model systems and in the cell.

Finally, as previously mentioned in the background section, all of our experimental observations [12,13,14,15] are difficult to understand under the dominant view of the cell. We found an appropriate model to unify/interpret them, which is grounded in the long-dismissed protoplasmic view of the cell and rigorously formalized in the association induction hypothesis [17,18,19]. The metabolic regulation of the membrane structure proposed in this article, therefore, raises debate about the role of lipids and their supramolecular structures in cells. For example, classical concepts such as permeability or compartmentalization may need to be reevaluated if we accept the view of the cell as a gel responsive to metabolic activity, where molecular crowding dominates the emergent features of the intracellular environment and (polarized) intracellular water emerges as an active component in living systems.

## 6. Materials and Methods

*Materials.* The aptamer switch probe used to measure intracellular ATP concentration was obtained from VBC Biotech (Vienna, Austria). 6-acetyl-2-dimethylaminonaphthalene (ACDAN) was purchased from Santa Cruz Biotechnology Inc. (Dallas, TX, USA). Mitotracker^TM^ red CMXRos, 6-lauroyl-2-dimethylaminonaphthalene (LAURDAN), and 9-diethylamino-5-benzo[a]phenoxazinone (Nile red) were from Thermo Fisher (Waltham, MA, USA). All other chemicals were purchased from Sigma-Aldrich (Munich, Germany). All measurements were performed in triplicate and the results were shown to be reproducible in independent measurements.

*Cell growth.* Cells of the yeast *Saccharomyces cerevisiae* diploid strain BY4743 wild type and isogenic mutants from the Euroscarf collection were grown and harvested as described previously [50]. The starved cells were suspended to a density of 10% (*w*/*v*) in 100 mM potassium phosphate buffer, pH 6.8, with or without 20 mM NaCl or in 100 mM sodium phosphate buffer, pH 6.8, and starved for 3 h at room temperature before use.

*Staining of cells with ACDAN, LAURDAN, Nile red, and MitoTracker red***.** Yeast cells (10% *w*/*v*) in 100 mM potassium phosphate buffer, pH 6.8, were incubated at 30 °C with 10 μM ACDAN, 20 μM LAURDAN, 5 μM Nile red, or 1 μM MitoTracker red for 1 h, washed twice, and finally re-suspended in the same buffer. For further details about the experiments see [12,13,15].

*Measurement of intracellular ATP***.** Measurements of intracellular ATP using an aptamer-based nanobiosensor were done essentially as described earlier [50]. The nanosensor consists of an approximately 30 nm polyacrylamide particle containing the aptamer switch probe BlackHole2-GTAGTAAGAACTAAAGTAAAAAAAAAATTAAAGTAGCCACGCTT-[CH2-CH2-O]36-TTACTAC-TexasRed with Alexa Fluor 388 dextran as the reference dye. The sensor was inserted into the cells by electroporation as described previously [51]. The sensor uses the ratio of the Texas Red fluorescence and Alexa Fluor 488 to determine the intracellular ATP concentration and can be calibrated in vitro. Here, the ATP concentration was estimated from a calibration curve constructed by measuring the fluorescence from the ATP sensor in mixtures of ATP and ADP where the total concentration of ATP plus ADP was 4 mM [51]. Temporal measurements of the fluorescence of the ATP nanosensor were made as the ratio of the emission at 605 with an excitation of 580 nm (Texas Red) over the emission at 520 nm with an excitation at 470 nm (Alexa Fluor 488) in a SPEX Fluorolog spectrofluorometer (Edison, NJ, USA) fitted with a temperature-controlled cuvette holder (Quantum Northwest, Liberty Lake, WA, USA). The sampling frequency was 1 Hz.

*Time resolved measurement of NADH, ADCAN, LAURDAN, Nile red, and Mitotracker red.* Measurements of NADH, ACDAN, LAURDAN, Nile red, and Mitotracker red in cuvette were made in a SPEX Fluorolog spectrofluorometer (Edison, NJ, USA) fitted with a temperature-controlled cuvette holder (Quantum Northwest, Liberty Lake, WA, USA). The temperature of the sample was maintained at 25 °C. NADH was excited at 366 nm, and its fluorescence emission was measured at 450 nm. ACDAN and LAURDAN were excited at 366 nm and the emissions were sampled at 440 nm. Nile red was excited at 520 nm and the emission was sampled at 600 nm. Mitotracker red was excited at 525 nm and emission was recorded at 605 nm. Sampling rates for all measurements was 1 Hz. For further details about ACDAN, LAURDAN, and Nile red experiments, see [12,13,15].

*Measurement of oscillations.* Yeast cells at 10% (*w*/*v*) in 100 mM phosphate buffer, pH 6.8, were added to a 1 mL cuvette mounted in the fluorometer under stirring conditions. Oscillations were induced by first adding 30 mM glucose, and 60 s later adding 5 mM KCN to the suspension. The sampling frequency was 1 Hz.

*Microscopy.* Fluorescence images of Mitotracker red were obtained in a Leica DMRE epifluorescence microscope equipped with a CoolLed illumination system (CoolLed, Andover, UK) a green/red filter cube (BP 515–560 nm, Dichroic 580 nm, LP590 nm) and a 100X Leica oil-emission objective (NA = 1.4).

## Figures and Tables

**Figure 1 biomolecules-09-00687-f001:**
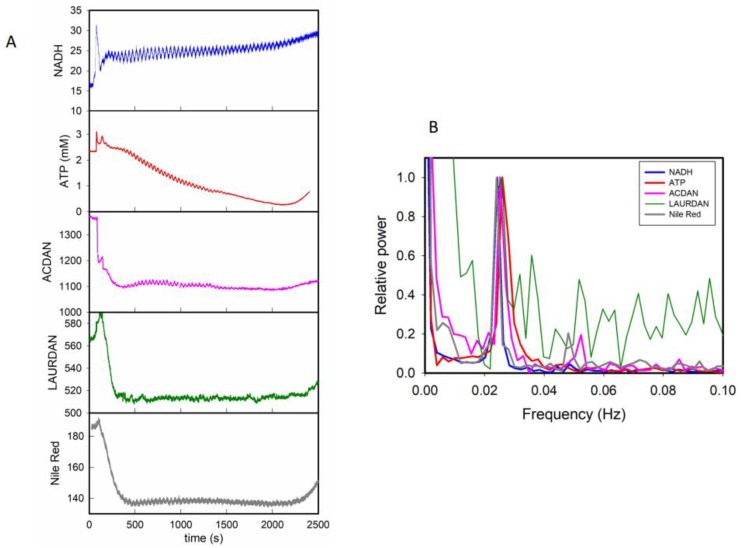
Oscillations in NADH, intracellular ATP, and the fluorescence of ACDAN, LAURDAN, and Nile red were synchronized. (**A**) Oscillations in NADH fluorescence, intracellular ATP concentration, ACDAN fluorescence, LAURDAN fluorescence, and Nile red fluorescence in a cell suspension with oscillating glycolysis. The cell suspension was 10% (*w*/*v*) *S. cerevisiae* BY4743 in 100 mM potassium phosphate buffer, pH 6.8, and temperature of 25 °C. Oscillations in glycolysis were induced at approximately t = 100 s by adding 30 mM glucose and 5 mM KCN to the cell suspension. (**B**) Power spectra of the oscillations in (**A**). Experimental details can be found in the Materials and Methods section.

**Figure 2 biomolecules-09-00687-f002:**
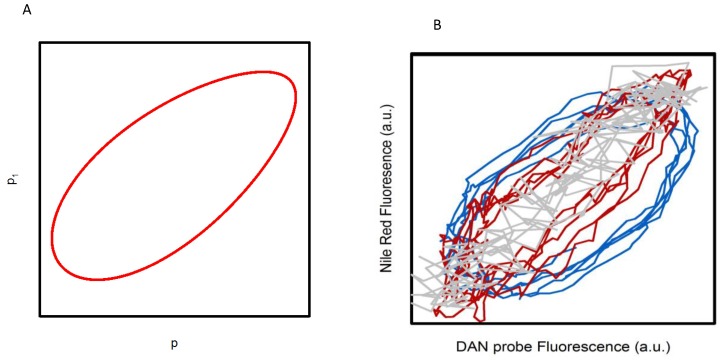
Oscillations in the polarization of water in aqueous environments with glycolytic oscillations may induce oscillations in (lipid-rich membranous) regions where no biochemical oscillations are expected to appear. (**A**) Theoretical phase plot of oscillations of polarization of water in a region with no glycolytic oscillations (p_1_) against the polarization of water in a region with glycolytic oscillations (p) generated by a Yang-Ling isotherm-based model of glycolytic oscillations. (**B**) Phase plot of Nile red fluorescence against the fluorescence of the three DAN probes: ACDAN (blue), PRODAN (red), and LAURDAN (grey). Reproduced from [13] and [15].

**Figure 3 biomolecules-09-00687-f003:**
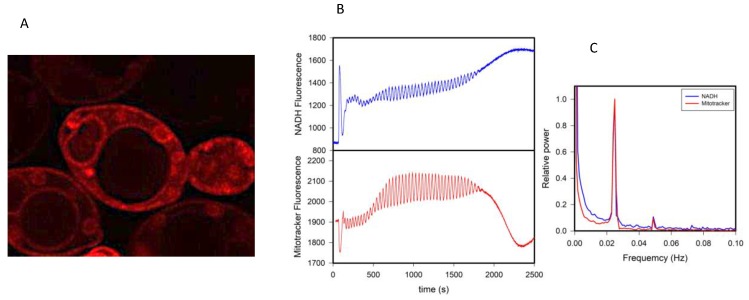
Membrane probes such as Mitotracker red may respond to changes other than membrane potential. (**A**) Fluorescence image of *S. cerevisiae* BY4743 cells stained with 1 μM Mitotracker red. (**B**) Oscillations in the fluorescence of NADH and Mitotracker red in a suspension of yeast cells where oscillations in glycolysis were induced by the addition of glucose and KCN as seen in Figure 1. (**C**) Power spectra of the oscillations in (**B**). Experimental details can be found in the Materials and Methods section.

**Figure 4 biomolecules-09-00687-f004:**
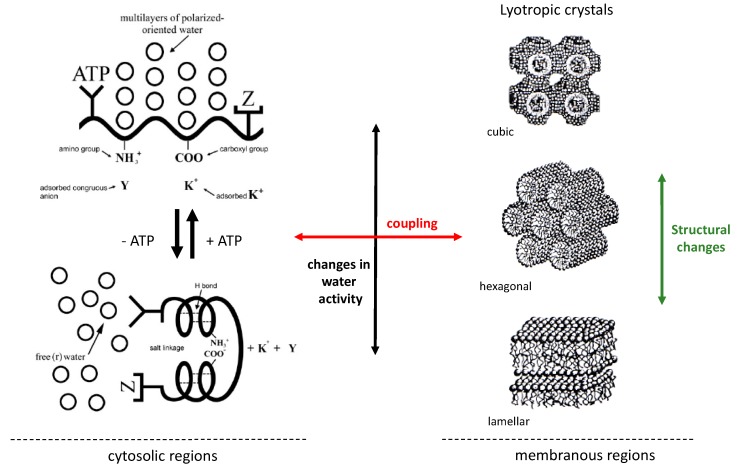
Sketch of the coupling between cytosolic and membranous regions. Changes in the activity of water caused by the polarization of intracellular water—via metabolic effects—can induce lyotropic regulation of the membrane structure with, for example, important curvature effects. For more details, see Section 4. Part of the figure is adapted from [19].

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
