# Peer review of "Coupled Response of Membrane Hydration with Oscillating Metabolism in Live Cells: An Alternative Way to Modulate Structural Aspects of Biological Membranes?"

_biomolecules, 2019, doi:10.3390/biom9110687_

Round 1

Reviewer 1 Report

Bagatolli, Stock and Olsen present an opinion article which is a mix of background, a review/repeat of their recent published work, new experimental data and suggestions for future work on the subject of membrane hydration and oscillating metabolism in cells. This article is written superbly and is interesting to someone from outside of this area of study as an introduction. It is a bit strange having a mix of data and review, and its primary intention appears to boost the profile of their recent studies but I think the format works and summarises the area well. I believe this article is suitable for publication although I do have some minor comments:

The authors make reference to their own groups work frequently, particularly references 13 and 15. So much so it at times feels as though this is the main aim of the article.

A lot of the results seem to be dependent upon the quality of small molecule fluorescent stains. The opinion could benefit from a brief discussion as to whether there are new dyes available or their quality. Presumably this could make a big impact in the area?

How do the authors feel aquaporins fit into their work?

Author Response

Response to Reviewer 1 Comments (our answer is indicated in red colour)

1) Bagatolli, Stock and Olsen present an opinion article which is a mix of background, a review/repeat of their recent published work, new experimental data and suggestions for future work on the subject of membrane hydration and oscillating metabolism in cells. This article is written superbly and is interesting to someone from outside of this area of study as an introduction.

We thank the reviewer for the positive overall statement.

2) It is a bit strange having a mix of data and review, and its primary intention appears to boost the profile of their recent studies but I think the format works and summarises the area well.

Thanks for the positive feedback. We decided to combine relevant data and review in the frame of our opinion article to stress the main message given in this work. It was not our intention to “boost” the profile of our recent papers; on the contrary, we are using the data reported to sustain a new hypothesis about the regulation of structural and dynamical aspects of membranes taking into account the acknowledged but seldom considered colloidal nature of the cell interior. In other words, we are incorporating effects on lipid structures caused by changes in intracellular water activity into the framework of the Association-Induction Hypothesis. This important aspect emerges as a new piece of information, which was not explored in our preceding papers or in the original formulation of the AIH.

3) I believe this article is suitable for publication although I do have some minor comments: The authors make reference to their own groups work frequently, particularly references 13 and 15. So much so it at times feels as though this is the main aim of the article.

We disagree with the reviewer. As mentioned above, the data presented in those papers is a crucial and original part of the whole data-set we employ to support the opinions we present.

4) A lot of the results seem to be dependent upon the quality of small molecule fluorescent stains. The opinion could benefit from a brief discussion as to whether there are new dyes available or their quality. Presumably this could make a big impact in the area?

We do not quite understand the point of the reviewer here. The quality of the probes only affects the results if impurities interfere with their properties, and we have no reason to doubt the purity of the preparations we use. There is a very extensive published literature on the characterization of these probes, by us and others; we invite the reviewer to check references 24 to 30 in the paper. In the particular case of LAURDAN, which is one of the more widely used probes to study lipid bilayers, there is a well-developed model of interpretation. In fact, the use of this probe has been consolidated as an established methodological tool (PubMED gives 579 hits of papers using LAURDAN as keyword). In addition, the results concerning the dynamical aspects of water obtained by us using the fluorescent dyes in cells are well in line with a wealth of independent experimental data (for example, Debye dielectric reorientation time, NMR rotational correlation time, vapor sorption at near-saturation vapor pressure, rotational diffusion coefficient from quasi-elastic neutron scattering, among others (see reference 19, pp78).

5) How do the authors feel aquaporins fit into their work?

At this time we are more concerned about the properties of the intracellular milieu, and the properties of intracellular water that are dynamically affected by an active metabolism, than how water moves in and out of the cell. In general, we feel that the proposed action of water channels presupposes that intracellular water is essentially akin to dilute solutions. Taking into account the dynamics of intracellular water, we think that the concept of water as a transported solute may require more detailed studies that are, as of now, very much outside the scope of our work.

Reviewer 2 Report

The study performed by Isabelle Kuan, et al., focused on “Coupled response of membrane hydration with 2 oscillating metabolisms in live cells: an alternative 3 way to modulate structural aspects of biological 4 membranes?.”

Overall, it is well conducted and well written.

I have some minor suggestions:

For better visibility of your work, illustrate figures or diagrams of the essential elements of your bibliographic reviews (as Ling’s Association-Induction Hypothesis) and results.

I declare no financial relationships with any organisations that might have an interest in the submitted review; no other relationships or activities that could appear to have influenced the submitted review.

Author Response

Response to Reviewer 2 Comments (our response are indicated in red color)

1) The study performed by Isabelle Kuan, et al., focused on “Coupled response of membrane hydration with 2 oscillating metabolisms in live cells: an alternative 3 way to modulate structural aspects of biological 4 membranes?.”

The reviewer made a typo concerning the name of authors and, maybe, a copy/paste error adding some numbers to the title of our article (editor, would you please amend?)

2) Overall, it is well conducted and well written.

Thanks.

3) I have some minor suggestions: For better visibility of your work, illustrate figures or diagrams of the essential elements of your bibliographic reviews (as Ling’s Association-Induction Hypothesis) and results.

Following the reviewer's suggestion, we have incorporated as Figure 4 in the new version of the manuscript an illustrative scheme of our hypothesis.